TOPICAL REVIEW

# A change of heart: Mechanisms of cardiac adaptation to acute and chronic hypoxia

Alexandra M. Williams[1,2] , Benjamin D. Levine[3] and Mike Stembridge[4]

[1] *Department of Cellular and Physiological Sciences, Faculty of Medicine, University of British Columbia, Vancouver, BC, Canada*
[2] *International Collaboration on Repair Discoveries, University of British Columbia, Vancouver, BC, Canada*
[3] *Institute for Exercise and Environmental Medicine, The University of Texas Southwestern Medical Center, Dallas, TX, USA*
[4] *Cardiff School of Sport and Health Sciences, Cardiff Metropolitan University, Cardiff, UK*

Handling Editors: Ian Forsythe & Andrew Holmes

The peer review history is available in the Supporting Information section of this article (https://doi.org/10.1113/JP281724#support-information-section).

**Dr Alex Williams** is a clinical cardiovascular physiologist, currently completing her postdoctoral fellowship at University of British Columbia's Faculty of Medicine and the International Collaboration on Repair Discoveries (ICORD). Her research is focused on the investigation of sex-based differences in the autonomic control of the heart, including the impacts of hypoxia on cardio-autonomic function. **Dr Benjamin Levine** is the founder and Director of the Institute for Exercise and Environmental Medicine (IEEM) at Texas Health Presbyterian Hospital Dallas. His global research interests centre on the adaptive capacity of the circulation in response to exercise training, deconditioning, aging, and environmental stimuli such as spaceflight and high altitude. **Dr Mike Stembridge** is a Reader in Cardiovascular and Environmental Physiology at Cardiff Metropolitan University. His research programme examines how the cardiovascular system adapts to meet the demands of exercise in a range of stressful environments and clinical conditions.

**Abstract** Over the last 100 years, high-altitude researchers have amassed a comprehensive understanding of the global cardiac responses to acute, prolonged and lifelong hypoxia. When lowlanders are exposed to hypoxia, the drop in arterial oxygen content demands an increase in cardiac output, which is facilitated by an elevated heart rate at the same time as ventricular volumes are maintained. As exposure is prolonged, haemoconcentration restores arterial oxygen content, whereas left ventricular filling and stroke volume are lowered as a result of a combination of reduced blood volume and hypoxic pulmonary vasoconstriction. Populations native to high-altitude, such as the Sherpa in Asia, exhibit unique lifelong or generational adaptations to hypoxia. For example, they have smaller left ventricular volumes compared to lowlanders despite having larger total blood volume. More recent investigations have begun to explore the mechanisms underlying such adaptive responses by combining novel imaging techniques with interventions that manipulate cardiac preload, afterload, and/or contractility. This work has revealed the contributions and interactions of (i) plasma volume constriction; (ii) sympathoexcitation; and (iii) hypoxic pulmonary vasoconstriction with respect to altering cardiac loading, or otherwise preserving or enhancing biventricular systolic and diastolic function even amongst high altitude natives with excessive erythrocytosis. Despite these advances, various areas of investigation remain understudied, including potential sex-related differences in response to high altitude. Collectively, the available evidence supports the conclusion that the human heart successfully adapts to hypoxia over the short- and long-term, without signs of myocardial dysfunction in healthy humans, except in very rare cases of maladaptation.

(Received 7 February 2022; accepted after revision 21 July 2022; first published online 5 August 2022)

**Corresponding author** Mike Stembridge: Cardiff School of Sport and Health Sciences, Cardiff Metropolitan University, Cardiff CF23 6XD, UK. Email: mstembridge@cardiffmet.ac.uk

**Abstract figure legend** Responses of key cardio-autonomic parameters to acute, prolonged, and lifelong hypoxia. Relative changes are plotted for lowlanders across the timelines of 0–12 h (left) and 1 day to 6 months (middle) and compared with observations in select populations residing at high altitude in Asia and South America (right). Where divergent adaptations occur, Andean and Sherpa data are represented by dashed and dotted lines, respectively. Generally, increasing exposure durations to hypoxia are characterized by substantial increases to sympathetic neural activation (SNA) and pulmonary arterial systolic pressure (PASP), with afterload-induced expansion of right ventricular end-diastolic area (RV EDA) and consequent reductions to left ventricular (LV) end-diastolic volume (EDV). Total blood volume (BV) is initially lowered with prolonged exposure as a result of plasma volume constriction but, over several months, returns to sea level values (or above) and is notably higher in high-dwelling Sherpa and Andeans compared to acclimatized lowlanders when expressed as relative to body size.

## Overview

Over 80 million humans permanently live at altitudes greater than 2500 m above sea level, of which 14.4 million reside at over 3500 m (Tremblay & Ainslie, 2021). Healthy humans who reside at low altitude also experience hypoxia during travel to mountainous regions for leisure, economic or military purposes, or as a consequence of pulmonary or cardiovascular disease (inherited or acquired) that impairs normal gas exchange at sea level. Whether short-term or lifelong, exposure to hypoxia requires adjustments to the cardiovascular system to maintain oxygen delivery in the face of decreased arterial oxygen saturation. Changes in cardiac structure and function in response to acute (0–12 h), prolonged (1 day to 6 months) and lifelong hypoxia have been well-described by over 100 years of scientific research (Lankford & Swenson, 2014). However, in some cases, the physio-logical mechanisms underpinning those cardiac responses have remained elusive. Herein, we review the mechanistic research that has advanced our understanding of how changes in preload, afterload and inotropic state influence cardiac structure and function specific to the duration of hypoxic exposure.

### Biventricular adjustments to acute hypoxic exposure

With exposure to acute hypoxia (i.e. 0–12 h), cardiac output becomes elevated primarily because of an increase in heart rate, whereas stroke volume is maintained (Vogel & Harris, 1967). This initial response is necessary to maintain oxygen delivery in the face of decreased oxygen saturation (e.g. $O_2$ delivery = arterial $O_2$ content × cardiac output). The increase in heart rate results from a combination of sympathetic activation

(Wolfel et al., 1998) and parasympathetic withdrawal (Koller et al., 1988; Wolfel et al., 1998). Stroke volume is generally maintained despite possible reductions to left ventricular (LV) end-diastolic volume (EDV) as a result of a proportionate reduction in end-systolic volume (ESV) (Baggish et al., 2013). Alongside the small reduction in LV EDV, right ventricular (RV) internal chamber diameter (Netzer et al., 2017) increases secondary to elevated pulmonary pressures via hypoxic pulmonary vasoconstriction (Motley et al., 1947). The onset of hypoxic pulmonary vasoconstriction occurs within 5 min of hypoxic exposure, followed by a more pronounced secondary increase over the subsequent 2 h (Talbot et al., 2005) and a third phase lasting for ∼8 h (Vejlstrup et al., 1997) (a dedicated review of hypoxic pulmonary vasoconstriction is provided in Swenson 2013). Hypoxic pulmonary vasoconstriction in acute hypoxia results in an increased RV chamber size, which is normalised when pulmonary pressure is reduced via sildenafil administration (Kjaergaard et al., 2007). By contrast to the pulmonary circulation, there is little change in pressure status across the systemic circulation because the sympathetically-mediated increases in vascular resistance are offset by local vasodilatory effects of hypoxia (Duplain et al., 1999; Hansen & Sander, 2004; Simpson et al., 2019, 2021). As such, increases to LV afterload do not appear to occur in acute hypoxia (Williams et al., 2019). Therefore, the increases and reductions to RV and LV dimensions, respectively, probably result from direct ventricular interaction secondary to hypoxic pulmonary vasoconstriction, albeit any such shifts appear to be relatively mild.

Global left and right ventricular function are preserved in acute hypoxia (Goebel et al., 2013; Huez et al., 2005; Suarez et al., 1987), with the maintenance of global ejection and filling accompanied by subtle changes in regional mechanical function. Over the last decade, a collection of studies has utilized speckle-tracking analysis of echocardiographic images to obtain indices of left ventricular mechanics in hypoxic environments. This technique allows for the assessment of regional function in multiple planes of myocardial deformation. For the purposes of this review, we focus on (i) LV twist as a marker of systolic function, defined as the net counter-directional rotation of the base and apex (Fig. 1*A*); (ii) strain, the deformation of the myocardium through the circumferential or longitudinal planes, expressed as percentage change from end-diastole (Fig. 1*B*); and (iii) the respective rates of these mechanics during both systole and diastole (Sengupta et al., 2008). With acute hypoxia, there is a rise in biventricular longitudinal strain and systolic strain rate (Goebel et al., 2013), alongside increased LV twist, circumferential strain (Dedobbeleer et al., 2013) and myocardial systolic tissue velocity (Huez et al., 2005). Provided that there are negligible changes to LV preload, these

mechanical responses are indicative of sympathetic activation, the majority of which return to sea level baseline under cardio-selective beta-adrenergic receptor (ß-AR) blockade (Dedobbeleer et al., 2013; Williams et al., 2019). Although the sympathoexcitatory effects of hypoxia on RV systolic function are less clear, sympathetic activation of the RV myocardium during acute hypoxia would be expected to be relatively comparable given the similarities between RV and LV nerve fibre (Kawano et al., 2003; Wharton et al., 1990) and adrenergic receptor densities (Chester & Barnett, 1995; Steinfath et al., 1992). However, unlike the LV, the RV is challenged in hypoxia by increased afterload secondary to hypoxic pulmonary vasoconstriction, which may counter RV inotropic activation and limit any increases to RV longitudinal strain (Huez et al., 2005; Kjaergaard et al., 2007). Indeed, when hypoxic pulmonary vasoconstriction is blunted via sildenafil administration, systolic tricuspid annular velocity, another marker of longitudinal RV function, is increased compared to baseline and hypoxia alone (Kjaergaard et al., 2007). Alterations to diastolic function during acute hypoxia may also be attributed to sympathetic activation. For example, although early diastolic filling blood velocities are maintained in both ventricles, late atrial filling velocities are elevated, as would be expected during sympathoexcitation (Huez et al., 2005). Likewise, LV untwisting velocity increases during acute isocapnic hypoxia and is restored to normoxic levels with ß$_1$-AR blockade (Williams et al., 2019). In summary, global and regional biventricular function are supported by sympathetic activation that counters RV afterload to ultimately maintain stroke volume. When coupled with increased heart rate, the preserved stroke volume allows for a greater cardiac output in acute hypoxia.

## Cardiac function during acclimatisation to high altitude

**Global systolic function.** During the initial days at high altitude (>3000 m), stroke volume begins to decline, whereas heart rate remains elevated, resulting in a cardiac output comparable to sea level both at rest and during exercise (Alexander et al., 1967; Klausen, 1966). The lowered stroke volume occurs with a reduction to EDV that is relatively greater than the reduction to ESV, with this being underpinned by multiple physiological mechanisms, with notable contributions from total blood volume and hypoxic pulmonary vasoconstriction (Fig. 2). Although it has been proposed that hypoxia directly impairs the myocardium, it should be considered that this hypothesis is based on reports of altered $Ca^{2+}$ reuptake in isolated myofibre preparations (Silverman et al., 1997) and depressed cardiac function in invasive animal studies that control heart rate and ß-adrenergic signalling

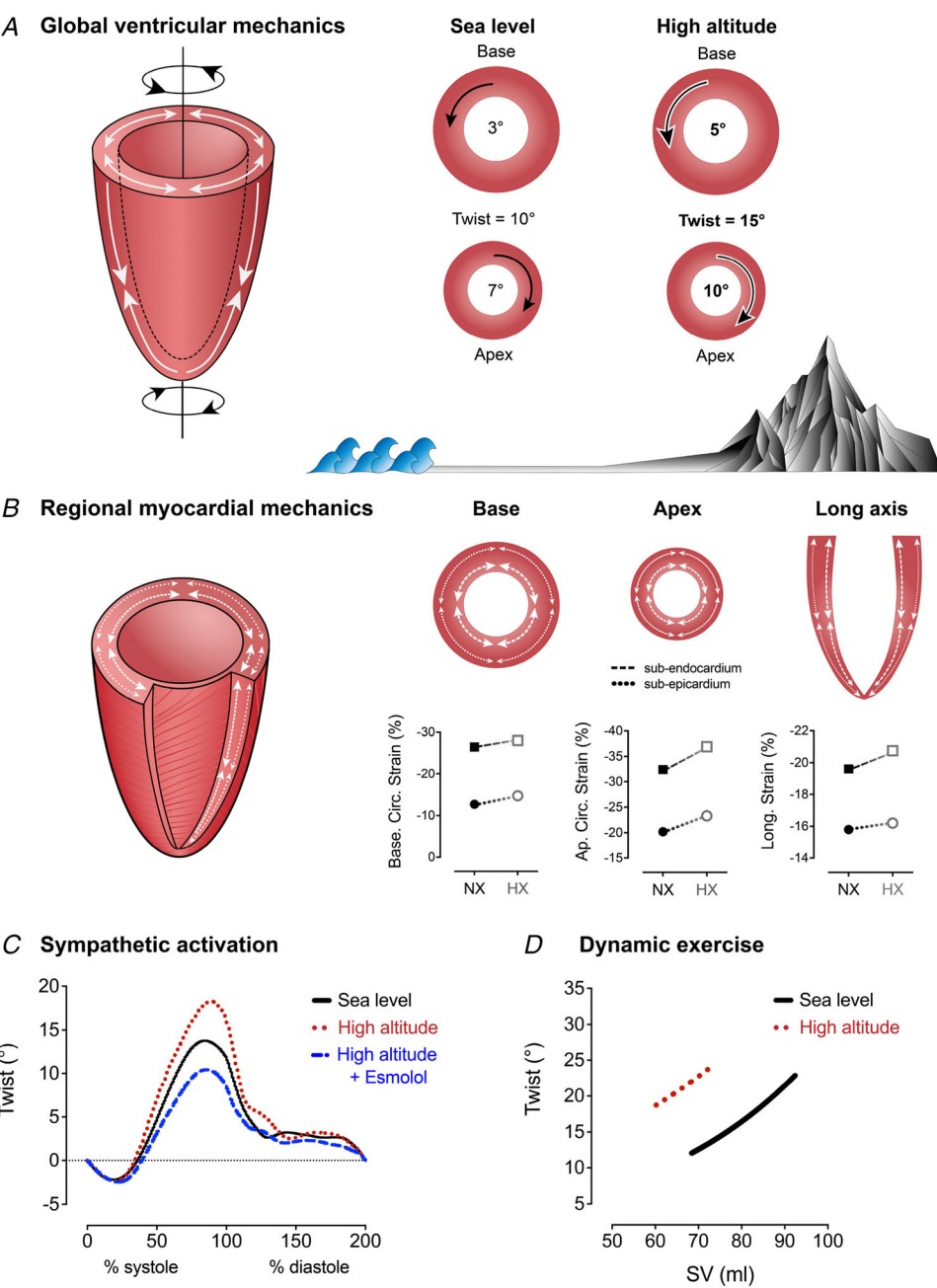

**Figure 1. Adjustments of global and regional ventricular mechanics with hypoxia, at rest and during exercise**

*A*, global left ventricular (LV) mechanics are elevated with both acute and prolonged hypoxia, where the LV twist is often augmented as a result of increased rotation at both the base and apex. *B*, recent work has assessed regional myocardial mechanics to determine whether subendocardial (square symbols, dashed line) *vs.* subepicardial mechanics (open circle symbols, dotted line) become impaired with hypoxia (HX) compared to normoxia (NX). There appear to be no indications of subendocardial dysfunction, as previously hypothesized, given the consistent increase in local strain in the basal (left), apical (middle) and longitudinal axes (right with acute and prolonged hypoxia. *C*, the increases to systolic mechanics are largely attributable to amplified sympathetic activation, as determined via administration of a cardiac-specific ß-adrenergic receptor blockade (ß$_1$-AR block). *D*, during exercise, LV twist mechanics become augmented from an elevated hypoxic baseline but reach a peak exercising twist similar to sea level. Representative data from Williams et al. (2019) and Stembridge et al. (2015).

(Tucker et al., 1976) and increased LV stiffness in dog models of hypoxic respiratory failure (Gomez & Mink, 1986). A classic study by Pool et al. (1966) in anaesthetized dogs found that indicators of LV contractile performance were either maintained or impaired with hypoxia, but only during complete autonomic blockade (hexamethonium ganglionic block plus ß-adrenergic block), which was not linked to myocardial ATP depletion. Autonomic blockade in hypoxia eventually led to LV failure in those animals, highlighting the critical role for the autonomic system in supporting healthy cardiac responses to hypoxaemia. However, the premise of myocardial dysfunction does not appear to be supported by studies in humans, where global cardiac function is clearly maintained; the seminal Operation Everest II studies (Reeves et al., 1987) demonstrated that stroke volume is preserved for a given right atrial pressure, even in simulated hypobaric hypoxia equivalent to the summit of Mt Everest (240 Torr). The same series of studies further reported that end-systolic elastance and contractility were also preserved during maximal exercise up to 7620 m (Suarez et al., 1987). Moreover, stroke volume is not restored with acute supplementary oxygen at 4300 m (Saltin et al., 1968), indicating that oxygen tension is not solely responsible for the reduced stroke volume at altitude, or at least that any such oxygen-dependent alteration to stroke volume

is not acutely reversible (Boussuges et al., 2000; Fowles & Hultgren, 1983; Reeves et al., 1987, 1990)

**Regional mechanical systolic function.** The maintenance of global systolic function with hypoxia is supported by regional adjustments, as highlighted by speckle-tracking studies conducted over the last decade. For example, longitudinal function assessed by LV strain is maintained following acclimatisation (Stembridge et al., 2014), and supported by similar observations from tissue Doppler measurements of longitudinal myocardial velocity (Huez et al., 2009). Additionally, LV twist is consistently reported to be elevated with high altitude acclimatization as a result of enhanced rotation of the LV apex and base (Dedobbeleer et al., 2013; Maufrais et al., 2016; Osculati et al., 2016; Stembridge et al., 2014; Stembridge, Ainslie et al., 2019; Williams et al., 2019). This twist or 'wringing' motion is generated by the LV's unique helical myofibre orientation (Fig. 1*A*). The elevated resting LV twist is most commonly interpreted as an adaptive response to maintain stroke volume via reducing end-systolic volume (Dedobbeleer et al., 2015; Maufrais et al., 2016; Stembridge et al., 2014). It has alternatively been proposed that elevations to LV twist result from (i) altered LV geometry as a result of reduced filling and (ii) subendocardial dysfunction (Osculati et al., 2015). Indeed, LV geometry has been identified as an independent predictor of LV twist (van Dalen et al., 2010) and LV mechanics are load-dependent (Kroeker et al., 1995; Weiner et al., 2010); however, LV twist still remains elevated when EDV is restored with saline infusion at high altitude (Stembridge, Ainslie et al., 2019). As such, the higher twist at altitude appears unrelated to changes in preload; rather, it is probably a result of increased contractility. The hypothesis for subendocardial dysfunction is derived from the concept that the inner myocardium is more susceptible to hypoxaemia and ischaemia, which would intensify the subepicardial dominance of LV twist. However, in a recent mechanistic investigation, Williams et al. (2019) applied region-specific imaging and found that subendocardial mechanics are preserved in both acute and prolonged hypoxia (Fig. 1*B*). The same study further revealed that increases to resting LV twist are blunted with ß$_1$-AR blockade following 3–6 days at 5050 m (Fig. 1*C*), suggesting that increased LV twist most probably results from hypoxia-induced sympathoexcitation, rather than subendocardial dysfunction (Williams et al., 2019). Furthermore, given that the studies examining myocardial mechanics are performed at rest in young, healthy individuals, subendocardial ischaemia and/or dysfunction are highly improbable given the substantial dilatory reserve of the coronary vasculature (Duncker & Bache, 2008). The elevated twist at rest may further influence cardiac function during exercise. Specifically,

**Sea level, normoxia**

2-4 mmHg 16-25 mmHg 6-12 mmHg

**Prolonged hypoxia**

↓ 0-2 mmHg ↑ 25-40 mmHg ↓ 3-5 mmHg

The Journal of
**Physiology**

**Figure 2. Alterations to cardiac pressures and dimensions with acclimatization or adaptation to hypoxia/high-altitude**
Left: normal health range of atrial and pulmonary arterial pressures, as well as ventricular-septal geometry, at sea level or long-term normoxia. Right: shifts in these parameters with prolonged or lifelong hypoxia. Here, atrial pressures are lowered as a result of reduced filling via lower blood volume, as well as hypoxic pulmonary vasoconstriction, which imposes a greater afterload on the right side of the heart, reduces right ventricular (RV) output and leads to RV expansion. Consequently, left ventricular (LV) end-diastolic volume and left-sided filling are reduced via direct ventricular interaction (septal shift) and series ventricular interaction (reduced RV output due to higher afterload). Figure based on reports from (Boussuges et al., 2000; Fowles & Hultgren, 1983; Reeves et al., 1987, 1990)

there is a clear left and upward shift in the relationship between LV twist and stroke volume during incremental exercise, where a ceiling in twist appears to occur for a lower stroke volume (Stembridge et al., 2015) (Fig. 1*D*). Therefore, although myocardial systolic function is either maintained or enhanced, a higher LV twist and lower ESV at rest may limit the mechanical reserve available during exercise following acclimatization to high altitude.

As a result of RV's complex semi-lunar geometry and the inherent limitations of echocardiography, relatively less is known about RV systolic function with high altitude acclimatization despite being faced with a more substantial afterload challenge compared to the left heart. Following the initial increase (i.e. 8–10 h) in pulmonary artery pressure with acute hypoxia (Dorrington et al., 1997), pulmonary pressures remain elevated with prolonged hypoxic exposure because of the lowered alveolar $P_{O_2}$ even following ventilatory acclimatization (West et al., 2007). Similar to acute hypoxia, RV longitudinal strain is generally unchanged from normoxia/sea level up to ~4300 m (Huez et al., 2009; Maufrais et al., 2016; Stembridge, Ainslie et al., 2019), although a decrement in RV longitudinal strain has been reported after ~10 days at 5050 m (Stembridge et al., 2014). To our knowledge, no studies have examined the RV under sympathetic blockade at high altitude; however, it has been found that partial reversal of hypoxic pulmonary vasoconstriction at high altitude does not appear to impact RV longitudinal strain as seen in acute hypoxia (Stembridge, Ainslie et al., 2019). It should be noted that the absence of altered RV function in that study may be attributable to a relatively small drop in pulmonary artery pressure (~4 mmHg) with sildenafil at moderate altitudes of 3800 m. By contrast to acute hypoxia where RV volume is increased, it is reasonable to speculate that RV function at high altitude may be depressed because of decreased filling pressure resulting in lower Frank–Starling mediated contractility, although this remains to be investigated.

**Diastolic function.** Given that LV EDV is lower at high altitude compared to sea level, and there is no evidence of overt systolic dysfunction, reduced ventricular filling probably underpins the decrease in stroke volume at high altitude. Similar to hypotheses for systolic dysfunction, it has been suggested that diastolic relaxation is impaired at high altitude (Holloway et al., 2011), although mechanistic evidence remains elusive. Conversely, most recent interrogations of LV relaxation indicate that LV early diastolic untwisting is enhanced (Dedobbeleer et al., 2015; Osculati et al., 2016; Stembridge et al., 2015), and this rapid recoil generates the atrioventricular and intraventricular pressure gradients for efficient diastolic filling (Notomi et al., 2007). As such, any hypothesised impairments

to diastolic myocardial function (e.g. stiffening, lower inotropy) do not appear to be supported by objective data, and hypoxia-induced shifts in LV diastolic measures are more probably a result of decreased total blood volume and return from the pulmonary circulation. Plasma volume constriction occurs during the first few days at high altitude (Ryan et al., 2014; Singh et al., 1990), in parallel with the decrease in LV EDV (Alexander et al., 1967). Grover et al. (1976) have demonstrated that stroke volume is unchanged from sea level to high altitude when plasma volume is maintained via $CO_2$ administration (3.77%) to avoid hypocapnic alkalosis and subsequent compartmental fluid shifts. In agreement, Siebenmann et al. (2013), Stembridge, Ainslie et al. (2019) and Gatterer et al. (2021) have all recently shown that acute restoration of plasma volume via saline infusion can restore stoke volume and ventricular filling at 3454–3800 m, although it should be noted that plasma volume expansion does not always augment LV stroke volume at higher elevations (~5000 m) (Alexander et al., 1967; Calbet, 2004). Therefore, although the decrease in circulating blood volume may indeed contribute to reduced ventricular filling at altitudes of ~3500 m, additional mechanisms may further contribute to this phenomenon at more extreme altitude.

Hypoxic pulmonary vasoconstriction produces a significant afterload challenge to the RV that can then lower LV filling via ventricular interactions (Fig. 2). First, increased RV afterload can reduce RV stoke volume, which in turn would lower LV filling via series ventricular interaction (Belenkie et al., 2001). Moreover, an increase in right-sided end-diastolic pressure and volume can lead to a septal shift towards the left side that would then augment LV transmural pressure and challenge LV filling via direct ventricular interaction (Belenkie et al., 2001). Both phenomena are known to restrict LV filling in pulmonary hypertension (Lumens et al., 2010) and septal flattening has been observed at 5050 m (Stembridge et al., 2014). Indeed, LV EDV is increased when pulmonary artery pressure is lowered with Sildenafil after 5–10 days at 3800 m (Stembridge, Ainslie et al., 2019). As such, the increases in pulmonary artery pressure at high altitude probably play a role in lowering LV filling, and this effect may become amplified at higher altitudes where the hypoxic pulmonary vasoconstriction response is greater.

## Lifelong high-altitude adaptation in Himalayan and Andean natives

An estimated 14.4 million people currently reside at altitudes of over 3500 m, with areas of Asia and South America accounting for the greatest absolute numbers (Tremblay & Ainslie, 2021). Following human migration from Africa 60 000 years ago, what we now consider

Tibetan and Sherpa populations settled at high altitude ~25 000 years ago. As the human migratory journey continued, ~10 000 years later, humans arrived in the Andean mountains of South America. These two key high altitude-dwelling populations have been studied extensively aiming to determine how humans adapt to lifelong hypoxic exposure. The journey of discovery was led by Peruvian scientists over 50 years ago, and the inspiring history has been reviewed previously (Reeves & Grover, 2005). In high altitude natives, elevated pulmonary pressures are not acutely reversible with oxygen supplementation and there is marked remodelling of smooth muscle around the pulmonary arteries and arterioles (Penaloza & Arias-Stella, 2007). However, pulmonary pressures can be lowered when the hypoxic stimulus is chronically removed via migration to sea level (Sime et al., 1971). As a consequence of the elevated pulmonary pressure, Andeans demonstrate mild right ventricular hypertrophy throughout the life span

(Penaloza & Arias-Stella, 2007). Himalayan Sherpa also demonstrate a smaller relative LV EDV compared to lowlanders, in both adults and children (Stembridge et al., 2014, 2016), necessitating a higher heart rate to achieve the required cardiac output. This lower stroke volume and higher heart rate appear to be a universal cardiac phenotype of humans at high altitude. However, unlike lowlanders where a lowered stroke volume largely results from a decreased blood volume, total circulating volume is significantly elevated in high altitude natives (Fig. 3). In Andeans, this is driven by substantial Hb mass expansion, whereas Sherpa demonstrate both a greater Hb mass and an expanded plasma volume compared to sea level residents (Claydon et al., 2004; Stembridge, Ainslie et al., 2019). Therefore, in high altitude natives, the lower LV EDV most probably results from elevated pulmonary artery pressure. The healthy cardiac phenotype in highlanders can therefore be summarised as exhibiting mild right ventricular

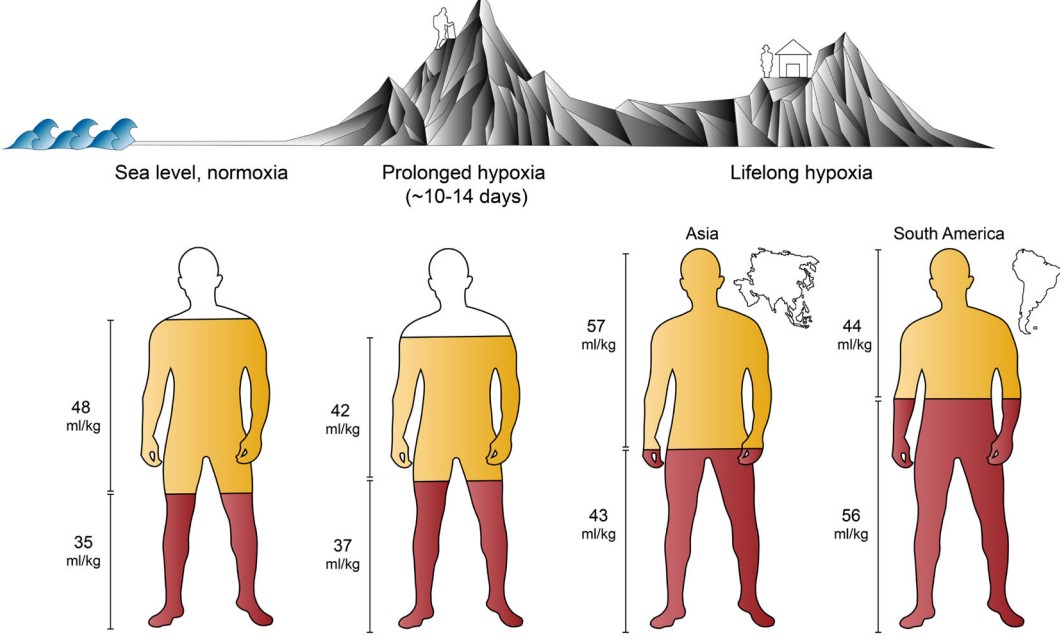

**Figure 3. Comparison of total blood volume and its components**
Comparison of total blood volume and its components amongst lowlander populations at sea level (left) and with prolonged hypoxia (middle), as well as populations residing at high altitude (right). Total fill indicates the relative proportions (i.e. scaled to body mass, ml/kg) of total blood volume, whereas red and yellow fills represent the relative red cell volume and plasma volume, respectively (representative data from Stembridge, Williams et al., 2019. In the earlier stage of prolonged hypoxia (~10–14 days), lowlanders initially experience plasma volume constriction, whereas the red cell volume generally remains unchanged, resulting in an increased haemoglobin concentration. As duration exposure increases from weeks to months, blood volume is gradually restored via erythrocytosis (Pugh, 1964; Reynafarje et al., 1959). Compared to lowlanders, high-dwelling populations have substantially larger blood volumes for their body size; however, Andean individuals often present with much higher proportions of red cell volumes, whereas Sherpa have haemoglobin concentrations comparable to those seen in acclimatized lowlanders (Stembridge, Williams et al., 2019).

hypertrophy secondary to hypoxic pulmonary vasoconstriction with reduced LV EDV and accompanying tachycardia. Unfortunately, a proportion of the population do not retain the healthy phenotype, and develop a condition termed 'chronic mountain sickness' (CMS) (Villafuerte & Corante, 2016).

CMS is a pathological condition that occurs most commonly in Andean high altitude natives (Penaloza & Arias-Stella, 2007) and Chinese Han immigrants (Wu et al., 1998) but less frequently in Tibetan populations (Pei et al., 1989). The condition is characterised by severe hypoxaemia relative to a given altitude and excessive erythrocytosis ($>21$ g $L^{-1}$ in males and $>19$ g $L^{-1}$ in females). These markers are accompanied by a range of debilitating symptoms including sleep disorders, headache, tinnitus and cognitive impairment (León-Velarde et al., 2010). Higher pulmonary artery pressures are also reported in CMS *vs.* healthy Andean natives (Maignan et al., 2009) and the pulmonary pressure response to exercise is greater in Chinese Han immigrants compared to Tibetan Natives (Yang et al., 1987). However, mathematical modelling (Vanderpool & Naeije, 2018) and haemodilution studies (Manier et al., 1988; Stembridge et al., 2021) have suggested that higher pulmonary pressures may be a consequence of increased viscosity as a result of polycythaemia, rather than exacerbated hypoxic pulmonary vasoconstriction *per se*. The higher pulmonary pressures exacerbate the RV dilatation that is commonly seen in Andeans (Maignan et al., 2009). Right heart failure is often assumed to be a common sequela of CMS, although there are very few data on its prevalence (Naeije & Dedobbeleer, 2013). Despite these morphological differences, LV and RV function are largely comparable between high altitude natives with and without CMS at rest (Pratali et al., 2013). Upon exertion, CMS patients experience an exaggerated increase in pulmonary artery pressure (Stuber & Scherrer, 2010) because of steep pressure–flow relationships. Healthy Andean natives can increase RV longitudinal function in response to submaximal exercise, whereas RV longitudinal velocity ($S'$) actually decreases in CMS patients (Pratali et al., 2013). Interestingly, the RV end-systolic pressure–area relationship, used as a non-invasive estimate of RV contractility, is preserved during submaximal exercise, indicating that CMS patients retain a contractile reserve. Therefore, although higher pulmonary pressures and RV dilatation are common features of CMS, ventricular–arterial coupling of the RV can be maintained, allowing the majority of high altitude residents to lead active lives (Vogel et al., 1963). The gradual and persistent nature of CMS contrasts reports of acute heart failure in Chinese Han immigrants. The condition is termed subacute mountain sickness (SMS), where hypoxic pulmonary vasoconstriction rapidly leads to right heart failure unless the patient descends (Anand et al., 1990). SMS was first reported in Indian soldiers deployed to high altitude areas as part of military service (Anand et al., 1990) and subsequently in paediatric patients who ascended to high altitude after residing at sea level (M et al., 2004; Sui et al., 1988). Anand et al. (1990) Interestingly, the primary mechanism of SMS is exaggerated sensitivity of the pulmonary vasculature with normal oxygen saturation and haemoglobin concentration relative to the altitude, which is in direct contrast to CMS where hypoventilation begins a cascade of life-long consequences. Differences in the aetiology of CMS and SMS highlight the race-specific nature of human (mal)adaptation to hypoxia.

## The male and female hearts in hypoxic environments

To date, the majority of studies examining the cardiovascular responses and physiological mechanisms involved in human adaptation to hypoxia have largely included either male-only or mixed-sex cohorts. In the earlier 20th century, a lack of female inclusion in hypoxia and altitude studies had been founded upon the historical deliberate exclusion of women from research teams and expeditions (Heggie, 2016; Tremblay et al., 2020). More recently, however, the substantial rise in female participation in mountaineering over the last several decades (Huey et al., 2020) has prompted increasing research into certain facets of female cardiopulmonary physiology and pregnancy in hypoxia (Beall et al., 2004; Drinkwater et al., 1982; Loeppky et al., 2001; Moore et al., 2004; Muza & Rock, 2001; Wagner et al., 1979, 1980; Zamudio et al., 2001). Although such research gaps in female physiology continue to be narrowed, remarkably little is known about the female heart in hypoxia, spanning the range of acclimation, acclimatization and lifelong adaptation. However, an increased focus on sex-related differences amongst cardiovascular physiologists has at least helped to identify key regulators of cardiac function amongst the female *vs.* male hearts (Bernacki et al., 2016; Burger et al., 2018; Jarvis et al., 2011; Lindenfeld et al., 2016; Salem et al., 2018; Simone et al., 1991; Tsuchimochi et al., 1995; Usselman et al., 2016; Williams et al., 2016, 2017, 2018), which may be important in the context of hypoxia and successful acclimatization and adaptation to high altitude environments.

To our knowledge, only a single study has examined sex-related differences in echocardiographic measures of ventricular structure and function with hypoxia. Boos et al. (2016) assessed young males and females at sea level and during acute normobaric hypoxia equivalent to 4800 m for 150 min, although they were unable to identify any sex-dependent effects of hypoxic exposure in their relatively small cohort ($n = 7$

per group). Although sex differences in the cardiac response to hypoxia have otherwise not been reported, several key factors may prove important in determining sex-dependent responses. Given the significant role of sympathoexcitation on cardiac function at altitude, divergent sympatho-vagal responses to hypoxia or general differences in cardio-autonomic control (i.e. receptor density, receptor sensitivity, autonomic reflex sensitivities, etc.) (Jarvis et al., 2011; Tsuchimochi et al., 1995; Usselman et al., 2016)) could contribute to different hypoxic cardiac responses in females *vs.* males. Unfortunately, the literature comparing cardio-autonomic control between the sexes during hypoxia is largely limited to HRV data, reporting inconsistent findings (Boos et al., 2017; Botek et al., 2018). Otherwise, a single study by Miller et al., 2019 has assessed MSNA and found no sex-related differences in the responses to very acute (5 min) hypoxia. Similarly, potential sex differences in hypoxic pulmonary vasoconstriction could impact the male and female hearts at altitude, although any such sex-specific data are limited: some mechanistic work in large animals suggests that potential sex-related differences in the NO-pathway regulation of pulmonary vascular function exist (Wijs-Meijler et al., 2017) and Fatemian et al. (2016) have reported exaggerated pulmonary arterial systolic pressure responses in females compared to males following 8 h of hypoxic exposure. Although earlier reports have broadly indicated that physiological 'performance' and changes to work capacity are comparable amongst healthy lowlander females and males in hypoxia (Drinkwater et al., 1979, 1982; Miles et al., 1980; Wagner et al., 1979), potential sex-related differences in the cardiopulmonary system and its autonomic control at high altitude largely remain to be explored.

**Regional geographical sex differences in chronic mountain sickness.** By contrast to the limited cardiopulmonary data, most of the sex-related literature at altitude has focused on the incidence CMS. The female 'hormonal milieu' has been hypothesized to provide some resistance to the negative sequelae of CMS (Ou et al., 1994), most notably in pre-menopausal women (Leon-Velarde et al., 2001; Moore, 2001). Specifically, the prevention of excessive erythrocytosis has been attributed to the protective role of oestrogen in females, with evidence from animal studies indicating that oestrogen can suppress the hypoxia-induced release of erythropoietin and the development of polycythaemia (Mirand & Gordon, 1966; Ou et al., 1994). Of note, Ou et al. (1994) have reported that ovariectomized rats exposed to 40 days of chronic hypoxia (simulated altitude 5500 m) have exaggerated cardiopulmonary adaptations, with higher haematocrit, pulmonary artery pressure,

RV weight and total heart volume compared to sham animals. It should be noted, however, that there appear to be notable regional divergences in the incidence of CMS amongst populations native to high altitude (Pooja et al., 2018), which could have implications for regional variations in sex differences related to CMS. For example, although CMS is more prevalent in females amongst Himalayan natives from the Spiti Valley, increased polycythaemia has also been reported in elderly males amongst Han, Tibetan and Mongolian populations (Okumiya et al., 2009). Furthermore, menopause has been identified as a risk factor for CMS and excessive erythrocytosis in Andeans (Gonzales et al., 2013; Heinrich et al., 2020).

## Cardiac contribution to exercise capacity in hypoxia

It is well-accepted that aerobic exercise performance is reduced with increasing altitude, with elevations as low as 500–900 m reducing performance in trained endurance athletes (Gore et al., 1996). At sea level, the heart is central to supporting high rates of convective oxygen transport to the skeletal muscle, largely setting the upper limit for maximal oxygen consumption ($\dot{V}_{O_2max}$) (Levine, 2008). However, in lowlanders at altitude, diffusional elements of oxygen transport are more influential on maximal oxygen uptake as a result of the decreased alveolar–arterial $P_{O_2}$ gradient (Wagner, 1996). Despite their diminished importance in oxygen transport, both blood volume and hypoxic pulmonary vasoconstriction have been investigated as potential contributors to decreased exercise capacity. Reducing hypoxic pulmonary vasoconstriction in acute hypoxia can restore ∼33% of the reduction in $\dot{V}_{O_2}$ from normoxia. A similar but smaller (∼10%) effect has been observed following pharmacological reversal of hypoxic pulmonary vasoconstriction at high altitude (Faoro et al., 2009; Faoro et al., 2009; Ghofrani et al., 2004; Hsu et al., 2006; Naeije et al., 2010) (Hsu et al., 2006; Richalet et al., 2005), although this has not always been effective (Faoro et al., 2007; Fischler et al., 2009; Stembridge, Ainslie et al., 2019). Any benefit is probably the result of improved gas exchange rather than the unloading of the right ventricle (Richalet et al., 2005). Restoring blood volume and haemoglobin concentration to sea level values has been shown to improve $\dot{V}_{O_2max}$ by 9% at 6000 m (Robach et al., 2000); however, the same effect has not been observed following dextran infusion following 9 weeks at 5260 m (Calbet, 2004) or saline infusion after 5–10 days at 3800 m (Stembridge, Ainslie et al., 2019). These interventional studies have failed to demonstrate a limiting role of the cardiovascular system during exercise at high altitude. Indeed, even recombinant human erythropoietin treatment will only enhance aerobic power at mild-moderate altitudes

(Young et al., 1996), highlighting a threshold ∼4000 m ($F_{IO_2} = 0.12$ to $0.13$) where the limited diffusion gradient renders any improvements in oxygen transport ineffective. Of note, Sherpa are able maintain their haemoglobin saturation at a higher level than their South American counterparts, and they also demonstrate a haemoglobin concentration near to sea level values (Beall, 2007) The 'normal' haemoglobin concentration is achieved by balancing an increase in Hb mass with plasma volume expansion (Stembridge, Williams et al., 2019). Regulation of blood volume in this manner affords Sherpa the ability to increase oxygen carrying capacity with a wider diffusion gradient and without the negative consequences of increased microvascular resistance and viscosity (Fan et al., 2010), which may facilitate enhanced blood flow (Erzurum et al., 2007; Gilbert-Kawai et al., 2017). Therefore, the importance of enhanced convective oxygen transport at high altitude may be dependent on a combination of adaptations that serve to increase overall oxygen utilisation.

## Summary

In conclusion, the human heart has considerable capacity to respond to hypoxic challenges, both over the acute and acclimatization periods for lowlanders and the lifelong or multigenerational hypoxic exposures experienced by populations residing at high altitudes. The current data suggest that the heart can effectively adjust to hypoxia-induced alterations in biventricular preload and afterload under hypoxia, predominantly via sympathoexcitation. Irrespective of duration of exposure, there is a pattern of RV dilatation and decreased LV volume that is attributable to the decrease in blood volume in lowlanders and the increase in pulmonary artery pressure across all time domains. In the vast majority of cases, this pattern of remodelling is physiological, although some detrimental effects of hypoxia on the heart are evident in a small proportion of CMS and SMS patients. Despite recent advancements in this field, key questions remain, including (i) which mechanisms allow certain populations like the Sherpa to balance an increase in haemoglobin mass with an expanded plasma volume; (ii) whether a greater haemoglobin mass carries significant functional benefits for high altitude natives; and (iii) how biological sex may modulate the cardio-pulmonary, autonomic and haematological responses to hypoxia. Understanding these key phenotypical differences will provide further clarity with respect to how different underlying factors, such as plasma volume regulation, autonomic control and sex hormones, contribute to the successful physiological adjustment of the heart and cardiopulmonary systems to low oxygen environments.

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

## Additional information

### Competing interests

The authors declare that they have no competing interests.

## Author contributions

All authors approved the final version of the manuscript submitted for publication. All persons designated as authors qualify for authorship, and all those who qualify for authorship are listed.

## Funding

No funding.

## Keywords

adaptation, altitude, blood volume, cardiac function, hypoxaemia, hypoxia, pulmonary hypoxic vasoconstriction, twist

## Supporting information

Additional supporting information can be found online in the Supporting Information section at the end of the HTML view of the article. Supporting information files available:

**Peer Review History**

