## [Peer Review History · The Journal of Physiology]

A Change of Heart: Cardiac Adaptation to Acute and Chronic Hypoxia.

Alexandra Mackenzie Williams, Benjamin D Levine, and Mike Stembridge
DOI: 10.1113/JP281724

Corresponding author(s): Mike Stembridge (mstembridge@cardiffmet.ac.uk)

The following individual(s) involved in review of this submission have agreed to reveal their identity: Christoph Siebenmann (Referee #1)

Review Timeline:

Submission Date:	07-Feb-2022
Editorial Decision:	15-Mar-2022
Revision Received:	06-Jul-2022
Accepted:	21-Jul-2022

Senior Editor: Ian Forsythe

Reviewing Editor: Andrew Holmes

Transaction Report:

Dear Mike,

Re: JP-TR-2022-281724 "A Change of Heart: Cardiac Adaptation to Acute and Chronic Hypoxia." by Alexandra Mackenzie Williams, Benjamin D Levine, and Mike Stemberidge

Thank you for submitting your Topical Review to The Journal of Physiology. It has been assessed by a Reviewing Editor and by 2 expert referees and I am pleased to tell you that it is considered to be acceptable for publication following satisfactory revision.

The reports are copied at the end of this email. Please address all of the points and incorporate all requested revisions, or explain in your Response to Referees why a change has not been made.

NEW POLICY: In order to improve the transparency of its peer review process The Journal of Physiology publishes online as supporting information the peer review history of all articles accepted for publication. Readers will have access to decision letters, including all Editors' comments and referee reports, for each version of the manuscript and any author responses to peer review comments. Referees can decide whether or not they wish to be named on the peer review history document.

I hope you will find the comments helpful and have no difficulty in revising your manuscript within 4 weeks.

Your revised manuscript should be submitted online using the links in Author Tasks Link Not Available. This link is to the Corresponding Author's own account, if this will cause any problems when submitting the revised version please contact us.

You should upload:

- A Word file of the complete text (including any Tables);
- An Abstract Figure, (with accompanying Legend in the article file)
- Each figure as a separate, high quality, file;
- A full Response to Referees;
- A copy of the manuscript with the changes highlighted.
- Author profile. A short biography (no more than 100 words for one author or 150 words in total for two authors) and a portrait photograph of the two leading authors on the paper. These should be uploaded, clearly labelled, with the manuscript submission. Any standard image format for the photograph is acceptable, but the resolution should be at least 300 dpi and preferably more.

- A 'Cover Art' file for consideration as the Issue's cover image;
- Appropriate Supporting Information (Video, audio or data set https://jp.msubmit.net/cgi-bin/main.plex?form_type=display_requirements#supp).

To create your 'Response to Referees' copy all the reports, including any comments from the Senior and Reviewing Editors into a Word, or similar, file and respond to each point in colour or CAPITALS. Upload this when you submit your revision.

I look forward to receiving your revised submission.

Best wishes,

Ian D. Forsythe
Deputy Editor-in-Chief
The Journal of Physiology
<https://jp.msubmit.net>
<http://jp.physoc.org>
The Physiological Society
Hodgkin Huxley House
30 Farringdon Lane
London, EC1R 3AW
UK
<http://www.physoc.org>
<http://journals.physoc.org>

EDITOR COMMENTS

Reviewing Editor:

Thank you very much for taking the time and effort to put together what everyone agrees is an interesting and insightful review. We are all in agreement that this work will be of great interest to the readers of Journal of Physiology. The work covers the topic thoroughly and gives insight into the most recent studies and potential future directions. The reviewers have pointed out some areas that could be considered in order to help with accessibility and reach, especially those who are not specialists in the field. I agree that some minor tweaks with the summary figure (detailed by the reviewers) would be especially helpful. Please can the authors check through as there are a few minor typos and words missing. Overall though this is great work.

Senior Editor:

Thanks for an interesting review. In your re-write, please consider how to make your article appeal to the widest audience; consider re-writing the abstract to increase the factual content, summarising your review and come to a clear conclusion in the last sentence (avoid saying further work is required in the abstract). Please consider whether an additional figure would help in providing background to your review topic.

REFEREE COMMENTS

Referee #1:

This paper reviews the effects of acute, extended and life-long exposure to high altitude/hypoxia on cardiac function, with special emphasis on findings made using echocardiography. The authors have extensive expertise in this topic and the paper will hence be of great interest to readers of this journal. I have some suggestions that may help to further improve the manuscript:

- I think that readers with an interest in hypoxia, but with limited experience in echocardiography, could get somewhat lost in the detailed discussion of the different ultrasound measures. Of course, these detailed discussions are important and should remain in the manuscript, however, a few more summarizing sentences throughout the text would be helpful for these readers.
- In the paragraph about acute hypoxia, the authors write that increased ventricular function secondary to sympathetic activation preserves SV in the face of increased right ventricular afterload. Later, they state that hypoxic pulmonary vasoconstriction contributes to the reduction in SV during extended hypoxia. While this is supported by the Stenbridge et al. 2019 data, it is not clear why RV afterload has this effect during extended, but not acute hypoxia. The summarizing figure illustrates that PASP increases during acute hypoxia and remains stable thereafter whereas sympathetic activity increases further from acute to extended hypoxia. As such, the heart should be even more capable of overcoming the increased RV afterload during extended hypoxia. Can the authors comment on this?
- Summarizing figure: I wonder whether the authors could include more variables? For readers unfamiliar with the topic, the SV and HR responses would be a valuable addition. Furthermore, why are only EDVs, but not ESVs included? Finally, I find the logarithmic scale not ideal - it is easy to e.g. misread that the biggest changes in SNA occur in acute hypoxia and that only minimal changes occur thereafter. Since all the presented changes go to only 100% (or slightly higher for the Andeans), an axis ranging to 1000% seems not needed.
- I would further recommend adding, maybe at the beginning of each duration of hypoxia (where the overall changes in SV and cardiac output are summarized), a clear statement about changes in EDV and ESV. Currently, this information is somewhat distributed among the text and figure legends, which can make it easy for readers to lose the overview.
- While the effects of pulmonary vasoconstriction and resulting right ventricular afterload are thoroughly discussed, the increase in systemic blood pressure that can occur during extended hypoxia and thus increase LV afterload is not addressed. Even if the authors think this does not play a role, I recommend briefly commenting on it.
- Hypoxia-induced increases in heart rate reduce diastolic filling times, which could contribute to reductions in EDV, particularly if filling pressures are low. Also here it would be interesting to know the authors' opinion.
- In the first paragraph about acclimatization to high altitude, the authors reason that hypoxia might directly impair myocardial function and that this could reduce SV. However, if such a direct effect existed, would it not be expected to primarily occur during acute hypoxia, not after extended exposure, where CaO₂ is normalized?
- Legend of figure 1: "atrial pressures are lowered due to reduced total blood volume" - in case of the left atrium, the reduced RV output (see next page) also contributes to this. Further, the legend states that values are red/blue (indicating

increases or reductions) which is not the case.

- Last sentence before figure 2: this implies that SV is maintained during extended hypoxia, which is not the case.

- "Cardiac contribution to exercise capacity in hypoxia": The authors write that pulmonary vasodilators do not benefit exercise in chronic hypoxia, but several studies do report a beneficial effect: PMID: 15289213; PMID: 15516532; PMID: 20378601. In the next sentence, the authors have missed to give the reference for the effect of PV expansion at altitude (PMID: 10904032).

- There are several typos and some sentences where words are missing. As there are no page or line numbers it is unfortunately difficult to indicate them, but I am sure the authors can find them.

Referee #2:

This is a very interesting review that highlights some recent advances in our understanding of cardiovascular responses to high altitude acclimatisation and genetic adaptation. I found the data on sex-dependent differences particularly interesting. I just have minor comments that I would like the authors to consider:

1. Page 6 - opening paragraph - suggest introducing situations where humans experience hypoxia (including timeframes - is the review just focussed on high-altitude?), and explain acclimatisation vs genetic adaptation
2. Page 6: "increase in right ventricular (RV) dimensions" - what dimension? Wall thickening? Chamber volume?
3. Page 6: remove "and" from "...in heart rate and while stroke volume.."
4. Page 7: "that counters RV afterload to ultimately maintains stroke volume" - maintains should be singular
5. Page 7: insert space after and before brackets "Kawano et al., 2003)and adrenergic receptor densities(Ste..."
6. Page 7: "o ultimately maintains stroke volume" - maintains should be singular
7. Page 8: "and depressed measures of cardiac..." doesn't make sense
8. Page 9: "end-diastolic volume is reduced due direct ventricular" - due to direct?
9. The conclusion is rather limited and only summarises a small amount of the content (e.g. didn't address sex-dependent differences). Suggest expanding here and adding some suggestions for future directions, highlighting the important areas that need to be addressed.

REQUIRED ITEMS:

-Your MS must include a complete "Additional information section" with the following 4 headings and content:

Competing Interests: A statement regarding competing interests. If there are no competing interests, a statement to this effect must be included. All authors should disclose any conflict of interest in accordance with journal policy.

Author contributions: Each author should take responsibility for a particular section of the study and have contributed to writing the paper. Acquisition of funding, administrative support or the collection of data alone does not justify authorship; these contributions to the study should be listed in the Acknowledgements. Additional information such as 'X and Y have contributed equally to this work' may be added as a footnote on the title page.

It must be stated that all authors approved the final version of the manuscript and that all persons designated as authors qualify for authorship, and all those who qualify for authorship are listed.

Funding: Authors must indicate all sources of funding, including grant numbers. If authors have not received funding, this must be stated.

It is the responsibility of authors funded by RCUK to adhere to their policy regarding funding sources and underlying research material. The policy requires funding information to be included within the acknowledgement section of a paper. Guidance on how to acknowledge funding information is provided by the Research Information Network. The policy also requires all research papers, if applicable, to include a statement on how any underlying research materials, such as data, samples or models, can be accessed. However, the policy does not require that the data must be made open. If there are considered to be good or compelling reasons to protect access to the data, for example commercial confidentiality or legitimate sensitivities around data derived from potentially identifiable human participants, these should be included in the statement.

Acknowledgements: Acknowledgements should be the minimum consistent with courtesy. The wording of acknowledgements of scientific assistance or advice must have been seen and approved by the persons concerned. This section should not include details of funding.

-Please upload separate high quality figure files via the submission form.

-Author profile(s) must be uploaded via the submission form. Authors should submit a short biography (no more than 100 words for one author or 150 words in total for two authors) and a portrait photograph of the two leading authors on the paper. These should be uploaded, clearly labelled, with the manuscript submission. Any standard image format for the photograph is acceptable, but the resolution should be at least 300 dpi and preferably more. A group photograph of all authors is also acceptable, providing the biography for the whole group does not exceed 150 words.

END OF COMMENTS

Confidential Review

07-Feb-2022

Response to Reviewers

Senior Editor:

Thanks for an interesting review. In your re-write, please consider how to make your article appeal to the widest audience; consider re-writing the abstract to increase the factual content, summarising your review and come to a clear conclusion in the last sentence (avoid saying further work is required in the abstract). Please consider whether an additional figure would help in providing background to your review topic.

Thank you for the helpful comments and advice in relation to our review. We have added in greater factual content to the abstract to better reflect the main body of the review, and have removed mention of future directions. We have considered the suggestion of an additional figure and appreciate the suggestion. We do however believe we have captured the key factors discussed in the paper in the abstract figure, and additional concepts in the figures through the body of the paper. If the editor feels strongly about a specific concept or display they would like to see in the paper we would be happy to discuss that further.

Reviewing Editor:

Thank you very much for taking the time and effort to put together what everyone agrees is an interesting and insightful review. We are all in agreement that this work will be of great interest to the readers of Journal of Physiology. The work covers the topic thoroughly and gives insight into the most recent studies and potential future directions. The reviewers have pointed out some areas that could be considered in order to help with accessibility and reach, especially those who are not specialists in the field. I agree that some minor tweaks with the summary figure (detailed by the reviewers) would be especially helpful. Please can the authors check through as there are a few minor typos and words missing. Overall though this is great work.

Thank you for your kind comments about our review. Please see below where we have tried to incorporate each of the reviewer's suggestions, and have provided a detailed response where we believe it was not in the best interests of the manuscript. Thank you for taking the time to read and review our manuscript.

REFEREE COMMENTS

Referee #1:

This paper reviews the effects of acute, extended and life-long exposure to high altitude/hypoxia on cardiac function, with special emphasis on findings made using echocardiography. The authors have extensive expertise in this topic and the paper will hence be of great interest to readers of this journal. I have some suggestions that may help to further improve the manuscript:

We thank you for taking the time to review our manuscript, and for your kind comments. Please see below details of how we have incorporated your suggestions.

- I think that readers with an interest in hypoxia, but with limited experience in echocardiography, could get somewhat lost in the detailed discussion of the different ultrasound measures. Of course, these detailed discussions are important and should remain in the manuscript, however, a few more summarizing sentences throughout the text would be helpful for these readers.

Thank you for this helpful suggestion. We accept that the terminology was a little heavy in places. We have now added a few explanatory words throughout to help the reader understand the aim/concept of the particular echocardiographic measure.

- In the paragraph about acute hypoxia, the authors write that increased ventricular function secondary to sympathetic activation preserves SV in the face of increased right ventricular afterload. Later, they state that hypoxic pulmonary vasoconstriction contributes to the reduction in SV during extended hypoxia. While this is supported by the Stembridge et al. 2019 data, it is not clear why RV afterload has this effect during extended, but not acute hypoxia. The summarizing figure illustrates that PASP increases during acute hypoxia and remains stable thereafter whereas sympathetic activity increases further from acute to extended hypoxia. As such, the heart should be even more capable of overcoming the increased RV afterload during extended hypoxia. Can the authors comment on this?

The reviewer makes a good point to which we do not have a definitive answer to. We highlight that during acute hypoxia, RV area increases slightly while stroke volume is maintained. If PASP is reduced via Sildenafil, no change in RV area occurs indicating that hypoxic pulmonary vasoconstriction does indeed influence RV structure and function. Perhaps the main difference when compared to prolonged exposure is the reduced filling pressures seen at high altitude (Reeves *et al.*, 1990), which will lower contractility mediated by the Frank-Starling effect. Therefore, whilst the PASP-induced increase in right-sided afterload will influence RV function in both the acute and prolonged settings, the effect on left heart function is amplified in the face of lowered blood volume and left-sided filling pressures. We provided additional details to articulate this on lines 242-245, and framed that this is a speculative discussion.

- Summarizing figure: I wonder whether the authors could include more variables? For readers unfamiliar with the topic, the SV and HR responses would be a valuable addition. Furthermore, why are only EDVs, but not ESVs included? Finally, I find the logarithmic scale not ideal - it is easy to e.g. misread that the biggest changes in SNA occur in acute hypoxia and that only minimal changes occur thereafter. Since all the presented changes go to only 100% (or slightly higher for the Andeans), an axis ranging to 1000% seems not needed.

We understand and agree the scaling is tricky to work with here. We appreciate that the log scale is not perfect, however when we used a normal interval scale, we essentially lose the effects that are seen for several factors as they are operating on such a relatively small portion of the larger

scale and they overlap. For that reason, we chose to use the scale we did to allow for the clear visualization of changes in LV EDV, blood volume and RV area. We also understand that it would be ideal to have multiple additional variables included on the graphic, but when added we felt they either cluttered the graphic considerably. We hope the reviewer can appreciate the figure as a general summary.

- I would further recommend adding, maybe at the beginning of each duration of hypoxia (where the overall changes in SV and cardiac output are summarized), a clear statement about changes in EDV and ESV. Currently, this information is somewhat distributed among the text and figure legends, which can make it easy for readers to lose the overview.

We appreciate this point and had tried to incorporate this information throughout each section, but appreciate that it did not come across clearly. To maintain the flow of each paragraph, we have kept the general structure but added in comment on EDV, ESV and SV in the first few lines of the sections.

- While the effects of pulmonary vasoconstriction and resulting right ventricular afterload are thoroughly discussed, the increase in systemic blood pressure that can occur during extended hypoxia and thus increase LV afterload is not addressed. Even if the authors think this does not play a role, I recommend briefly commenting on it.

It is indeed relevant to consider the potential alterations to LV afterload. In a recent study we found that despite the increase in SBP, end-systolic wall stress, an index of LV afterload, was not increased with acute hypoxia or at 5050m altitude compared to sea level likely due to the relative reductions in LV filling. We have included this point in lines 134-135.

- Hypoxia-induced increases in heart rate reduce diastolic filling times, which could contribute to reductions in EDV, particularly if filling pressures are low. Also here it would be interesting to know the authors' opinion.

Whilst exercising heart rates may compromise diastolic filling due to the shortened time period available, the relatively minor increase in heart rate at rest (~10bpm) at high altitude is unlikely to influence filling. This has been demonstrated in a canine model (Weisfeldt *et al.*, 1978) where filling was preserved until the R-R interval was $< 3.5 \times \tau$ (heart rates of 170-200 bpm). We agree that the shorter filling times combined with lower filling pressures may decrease filling during exercise, but rarely do lowlanders achieve true maximal heart rates due to heightened parasympathetic neural activity (Boushel *et al.*, 2001). Therefore, we do not believe that filling time will adversely affect ventricular filling at high altitude.

- In the first paragraph about acclimatization to high altitude, the authors reason that hypoxia might directly impair myocardial function and that this could reduce SV. However, if such a direct effect existed, would it not be expected to primarily occur during acute hypoxia, not after extended exposure, where CaO₂ is normalized?

We apologize if this read as a statement by the authors. We were aiming to outline the various hypotheses related to myocardial function/malfunction with hypoxia that have been presented in the literature, one of which being that hypoxia could impair myocardial function. We do not believe this to be the case and have provided evidence later in the paragraph / section to show that this concept does not seem to be supported in the human literature. We have added to that paragraph to clarify how we have presented the data that support this concept in vitro / in animal models, but not in integrative human research.

- Legend of figure 1: "atrial pressures are lowered due to reduced total blood volume" - in case of the left atrium, the reduced RV output (see next page) also contributes to this.

Further, the legend states that values are red/blue (indicating increases or reductions) which is not the case.

- Last sentence before figure 2: this implies that SV is maintained during extended hypoxia, which is not the case.

Both points have been addressed in the figure legend.

- "Cardiac contribution to exercise capacity in hypoxia": The authors write that pulmonary vasodilators do not benefit exercise in chronic hypoxia, but several studies do report a beneficial effect: PMID: 15289213; PMID: 15516532; PMID: 20378601. In the next sentence, the authors have missed to give the reference for the effect of PV expansion at altitude (PMID: 10904032).

Thank you for highlighting these works. We have now incorporated these into the section and highlighted that pulmonary vasodilators have been shown to improve performance in some but not all cases, and where they have, it was likely due to improved gas exchange. We have also added the (Robach *et al.*, 2000) reference that was missing.

- There are several typos and some sentences where words are missing. As there are no page or line numbers it is unfortunately difficult to indicate them, but I am sure the authors can find them.

We have read through the manuscript closely in an effort to remedy any typos or missing words.

Referee #2:

This is a very interesting review that highlights some recent advances in our understanding of cardiovascular responses to high altitude acclimatisation and genetic adaptation. I found the data on sex-dependent differences particularly interesting. I just have minor comments that I would like the authors to consider:

We thank the reviewer for their careful consideration of our manuscript. Please see below where we have systematically addressed each of the concerns raised.

1. Page 6 - opening paragraph - suggest introducing situations where humans experience hypoxia (including timeframes - is the review just focused on high-altitude?), and explain acclimatisation vs genetic adaptation

We thank the reviewer for this suggestion, and agree that a little context in the introduction will help set the scene for the review. We have added sentences detailing where people live at high altitude, and when/where lowlanders will experience hypoxia.

2. Page 6: "increase in right ventricular (RV) dimensions" - what dimension? Wall thickening? Chamber volume?

We have now amended this section to refer specifically to internal chamber diameter.

3. Page 6: remove "and" from ". in heart rate and while stroke volume.."

This has been changed.

4. Page 7: "that counters RV afterload to ultimately maintains stroke volume" - maintains should be singular

This has been changed.

5. Page 7: insert space after and before brackets "Kawano et al., 2003)and adrenergic receptor densities(Ste "

This has been changed.

6. Page 7: "o ultimately maintains stroke volume" - maintains should be singular

This has been changed.

7. Page 8: "and depressed measures of cardiac. " doesn't make sense

This has been changed to remove the word "measures" as this was unnecessary.

8. Page 9: "end-diastolic volume is reduced due direct ventricular" - due to direct?

Thank you for catching this mistake- this has now been changed.

9. The conclusion is rather limited and only summarises a small amount of the content (e.g. didn't address sex-dependent differences). Suggest expanding here and adding some suggestions for future directions, highlighting the important areas that need to be addressed.

In line with the reviewer's advice, we have expanded the discussion to include critical findings from each section and have elaborated on future directions for the field to highlight what is still unknown. The concluding paragraph has been updated accordingly to include context on the key questions to still be addressed in the field, and reiterate the context of the discussion surrounding the acute, prolonged and lifelong cardiopulmonary adjustments to hypoxia.

References

- Boushel R, Calbet JAL, Rådegran G, Sondergaard H, Wagner PD & Saltin B (2001). Parasympathetic Neural Activity Accounts for the Lowering of Exercise Heart Rate at High Altitude. *Circulation* 104, 1785–1791.
- Reeves JT, Groves BM, Cymerman A, Sutton JR, Wagner PD, Turkevich D & Houston CS (1990). Operation Everest II: cardiac filling pressures during cycle exercise at sea level. *Respiration Physiology* 80, 147–154.
- Robach P, Déchaux M, Jarrot S, Vaysse J, Schneider JC, Mason NP, Herry JP, Gardette B & Richalet JP (2000). Operation Everest III: role of plasma volume expansion on VO₂(max) during prolonged high-altitude exposure. *J Appl Physiol* 89, 29–37.
- Weisfeldt ML, Frederiksen JW, Yin FCP & Weiss JL (1978). Evidence of Incomplete Left Ventricular Relaxation in the Dog. *J Clin Invest* 62, 1296–1302.

----- REQUIRED ITEMS:

-Your MS must include a complete "Additional information section" with the following 4 headings and content:

Competing Interests: A statement regarding competing interests. If there are no competing interests, a statement to this effect must be included. All authors should disclose any conflict of interest in accordance with journal policy.

Author contributions: Each author should take responsibility for a particular section of the study and have contributed to writing the paper. Acquisition of funding, administrative support or the collection of data alone does not justify authorship; these contributions to the study should be listed in the Acknowledgements. Additional information such as 'X and Y have contributed equally to this work' may be added as a footnote on the title page.

It must be stated that all authors approved the final version of the manuscript and that all persons designated as authors qualify for authorship, and all those who qualify for authorship are listed.

Funding: Authors must indicate all sources of funding, including grant numbers. If authors have not received funding, this must be stated.

It is the responsibility of authors funded by RCUK to adhere to their policy regarding funding sources and underlying research material. The policy requires funding information to be included within the acknowledgement section of a paper. Guidance on how to acknowledge funding information is provided by the Research Information Network. The policy also requires all research papers, if applicable, to include a statement on how any underlying research materials, such as data, samples or models, can be accessed. However, the policy does not require that the data must be made open. If there are considered to be good or compelling reasons to protect access to the data, for example commercial confidentiality or legitimate sensitivities around data derived from potentially identifiable human participants, these should be included in the statement.

Acknowledgements: Acknowledgements should be the minimum consistent with courtesy. The wording of acknowledgements of scientific assistance or advice must have been seen and approved by the persons concerned. This section should not include details of funding.

Dear Mike,

Re: JP-TR-2022-281724R1 "A Change of Heart: Cardiac Adaptation to Acute and Chronic Hypoxia." by Alexandra Mackenzie Williams, Benjamin D Levine, and Mike Stemberge

I am pleased to tell you that your Topical Review article has been accepted for publication in The Journal of Physiology, subject to any modifications to the text that may be required by the Journal Office to conform to House rules.

NEW POLICY: In order to improve the transparency of its peer review process The Journal of Physiology publishes online as supporting information the peer review history of all articles accepted for publication. Readers will have access to decision letters, including all Editors' comments and referee reports, for each version of the manuscript and any author responses to peer review comments. Referees can decide whether or not they wish to be named on the peer review history document.

The last Word version of the paper submitted will be used by the Production Editors to prepare your proof. When this is ready you will receive an email containing a link to Wiley's Online Proofing System. The proof should be checked and corrected as quickly as possible.

All queries at proof stage should be sent to tjp@wiley.com

The accepted version of the manuscript will be published online, prior to copy editing in the Accepted Articles section.

Are you on Twitter? Once your paper is online, why not share your achievement with your followers. Please tag The Journal (@jphysiol) in any tweets and we will share your accepted paper with our 22,000+ followers!

Best wishes,

Ian D. Forsythe
Deputy Editor-in-Chief
The Journal of Physiology
<https://jp.msubmit.net>
<http://jp.physoc.org>
The Physiological Society
Hodgkin Huxley House
30 Farringdon Lane
London, EC1R 3AW
UK
<http://www.physoc.org>
<http://journals.physoc.org>

*** IMPORTANT NOTICE ABOUT OPEN ACCESS ***

To assist authors whose funding agencies mandate public access to published research findings sooner than 12 months after publication The Journal of Physiology allows authors to pay an open access (OA) fee to have their papers made freely available immediately on publication.

You will receive an email from Wiley with details on how to register or log-in to Wiley Authors Services where you will be able to place an OnlineOpen order.

You can check if you funder or institution has a Wiley Open Access Account here <https://authorservices.wiley.com/author-resources/Journal-Authors/licensing-and-open-access/open-access/author-compliance-tool.html>

Your article will be made Open Access upon publication, or as soon as payment is received.

If you wish to put your paper on an OA website such as PMC or UKPMC or your institutional repository within 12 months of publication you must pay the open access fee, which covers the cost of publication.

OnlineOpen articles are deposited in PubMed Central (PMC) and PMC mirror sites. Authors of OnlineOpen articles are permitted to post the final, published PDF of their article on a website, institutional repository, or other free public server, immediately on publication.

Note to NIH-funded authors: The Journal of Physiology is published on PMC 12 months after publication, NIH-funded authors DO NOT NEED to pay to publish and DO NOT NEED to post their accepted papers on PMC.

EDITOR COMMENTS

Reviewing Editor:

Thank you for addressing the majority of the comments made by the reviewers. There are just a couple of minor points raised by the reviewers that need clarification.

Senior Editor:

Thank you for a thorough revision. The article is accepted, but please feel free to amend the text to clarify the minor points made by Ref 1 in the proofs.

Congratulations on an interesting review.

REFEREE COMMENTS

Referee #1:

I thank the authors for considering my comments. I have two minor additional comments on the authors responses to my previous report:

- Systemic blood pressure and LV afterload: The authors write that they have addressed this on line 134-135. I cannot find it there but find an insertion on line 143 of the version highlighting changes - is this what they meant? This paragraph refers to acute hypoxia where, as the authors state, the vasoconstrictive effects of sympathoactivation are counteracted by local vasodilatory mechanisms so that increases in LV afterload are indeed not expected. However, I think this point would be more relevant to address in the context chronic hypoxia, where increases in systemic blood pressure are expected (e.g. 10.1113/jphysiol.2003.045112), making increases in LV afterload more like.

- Heart rate and filling time: I agree with the authors' explanation, although it should be considered that during submaximal exercise the increase in heart rate induced by chronic hypoxia can be much larger than 10 bpm (e.g. 10.1152/ajpregu.00156.2002). For completeness, I would suggest including the authors' rationale against a role of filling times in the manuscript.

Referee #2:

All of my comments have been addressed.

1st Confidential Review

06-Jul-2022